# Ten Questions on Using Lung Ultrasonography to Diagnose and Manage Pneumonia in Hospital-at-Home Model: Part III—Synchronicity and Foresight

**DOI:** 10.3390/diagnostics16020192

**Published:** 2026-01-07

**Authors:** Nin-Chieh Hsu, Yu-Feng Lin, Hung-Bin Tsai, Charles Liao, Chia-Hao Hsu

**Affiliations:** 1Department of Internal Medicine, College of Medicine, National Taiwan University, Taipei 100225, Taiwan; 018519@ntuh.gov.tw (N.-C.H.);; 2Division of Hospital Medicine, Department of Internal Medicine, Taipei City Hospital Zhongxing Branch, Taipei 103212, Taiwan; 3Taiwan Association of Hospital Medicine, Taipei 100225, Taiwan; 4Department of Internal Medicine, Stanford University School of Medicine, Stanford, CA 94305, USA; 5Department of Orthopedics, Kaohsiung Medical University Hospital, Kaohsiung 807378, Taiwan; 6College of Medicine, Kaohsiung Medical University, Kaohsiung 807378, Taiwan

**Keywords:** pneumonia, point-of-care, ultrasonography, hospital-at-home, diagnosis, treatment, consolidation, air bronchogram

## Abstract

The hospital-at-home (HaH) model delivers hospital-level care to patients in their homes, with point-of-care ultrasonography (PoCUS) serving as a cornerstone diagnostic tool for respiratory illnesses such as pneumonia. This review—the third in a series—addresses the prognostic, synchronous, and potential overdiagnostic concerns of lung ultrasound (LUS) in managing pneumonia within HaH settings. LUS offers advantages of safety and repeatability, allowing clinicians to identify “red flag” sonographic findings that signal complicated or severe disease, including pleural line abnormalities, fluid bronchograms, absent Doppler perfusion, or poor diaphragmatic motion. Serial LUS examinations correlate closely with clinical recovery, showing progressive resolution of consolidations, B-lines, and pleural effusions, and thus provide a non-invasive method for monitoring therapeutic response. Compared with chest radiography, LUS demonstrates superior sensitivity in detecting pneumonia, pleural effusion, and interstitial syndromes across pediatric and adult populations. However, specificity may decline in tuberculosis-endemic or obese populations due to technical limitations and overlapping imaging patterns. Overdiagnosis remains a concern, as highly sensitive ultrasonography may identify minor or clinically irrelevant lesions, potentially leading to overtreatment. To mitigate this, PoCUS should be applied in parallel with conventional diagnostics and integrated into comprehensive clinical assessment. Standardized training, multi-zone scanning protocols, and structured image acquisition are recommended to improve reproducibility and inter-operator consistency.

## 1. Introduction

The hospital-at-home (HaH) model provides hospital-grade acute care in the patient’s home, and has been adopted in Western and Eastern countries [1,2,3,4]. HaH has proven effective in managing a range of acute medical conditions—including pneumonia, heart failure, COPD, urinary tract infections, and skin and soft tissue infections—by delivering timely, hospital-level care without the need for admission [4,5,6].

Successful HaH care hinges on deploying point-of-care services such as mobile diagnostics, real-time remote monitoring, intravenous treatments, and virtual consultations to replicate hospital capabilities at home [6,7]. Among available diagnostic tools, point-of-care ultrasonography (PoCUS) plays a central role [8]. While PoCUS complements other imaging modalities in hospital settings, it often functions as the primary—or even sole—diagnostic tool in HaH care [8,9,10,11,12,13].

Evidence supports the use of PoCUS in managing COVID-19 pneumonia, guiding the development of comparable home-based care strategies [14,15] for disease preparedness and mitigating health inequality [16,17]. Previously, we reviewed the techniques and key sonographic patterns for diagnosing pneumonia using PoCUS in home care settings and highlighted essential confounders and pathological mimickers that must be recognized (Table 1) [18,19]. This review further explores key questions regarding the prognostic value, synchronicity, and unexpected benefits of pneumonia management using PoCUS within the HaH model.

## 2. Question 7: Do Initial Ultrasound Findings Associated with Pneumonia Hold Prognostic Value?

During the treatment of infectious diseases, some imaging modalities are often used as a one-time cross-sectional assessment. For pneumonia, a typical one-time cross-sectional imaging is computed tomography (CT). A single-center study compared patients who were diagnosed with community-acquired pneumonia (CAP) using chest x-ray and CT in the emergency department. With a non-randomized retrospective analysis, it found that CT improved diagnosis consistency for CAP, with a trend for lower hospital length-of-stay around 2 days, but did not affect ICU admission and in-hospital mortality [20].

PoCUS offers advantages in safety and repeatability that are not typically achievable with CT. However, it is often performed only once during the entire treatment course—typically at enrollment—serving primarily as a triage tool to assess whether a pneumonia patient is suitable for out-of-hospital care. The additional prognostic insights provided by PoCUS are therefore essential to avoid assigning complex or difficult-to-treat pneumonia cases to home-based care.

PoCUS performed early in the course of pneumonia may provide valuable information regarding disease severity and the potential for complications. Table 2 summarizes red flag signs of LUS patterns which may be seen in patients with pneumonia.

### 2.1. Red Flag Signs Related to ARDS

Several LUS signs, based on previous studies, indicate a more severe pneumonia, such as acute lung injury or acute respiratory distress syndrome (ARDS). A landmark study sought to differentiate acute pulmonary edema (APE) from ARDS by LUS. Although B-lines (alveolar-interstitial syndrome) prevailed 100% in both APE and ARDS, absence or reduction of the pleural gliding (sliding) was consistently reported in ARDS and rarely observed in APE. ‘Spared areas’ within confluent B-lines were observed in 100% of patients with ARDS and in 0% of patients with APE [21]. In addition, pleural line abnormalities, including irregularity or thickening, were observed in 100% of patients with acute lung injury/ARDS (Figure 1). These signs are classified as level-B evidence, with strong recommendations in a landmark LUS guideline [22].

### 2.2. Red Flag Signs Related to Complicated Pneumonia

Fluid bronchogram is originally described as a sign on CT [23]. It is also described in post-obstructive pneumonia in ultrasonography, identified as anechoic tubular structures with hyperechoic walls but without color Doppler signals (Figure 2) [24,25].

Post-obstructive pneumonia with a fluid bronchogram usually reflects complete bronchial obstruction, making the consolidation refractory to antibiotic therapy alone.

The ultrasonographic appearance of pneumonia in children can be used for adults [22]. A study investigating pediatric hospitalized patients found that children with an uncomplicated CAP presented an air, arboriform, superficial and dynamic bronchogram, as opposed to complicated CAP, which had an air and liquid bronchogram, deep, fixed [26]. Another pediatric study reported that fluid bronchogram, multifocal involvement, and pleural effusion were associated with adverse outcomes, including longer hospital stay, ICU admission, and tube thoracotomy in hospitalized CAP children [27].

### 2.3. Red Flag Signs Related to Necrotizing Pneumonia

A study retrospectively reviewed 236 children with CAP. The perfusion of subpleural consolidation was classified into normal perfusion (homogenously distributed tree-like vascularity), decreased perfusion (less than 50% of an area with typical tree-like vascularity), and poor perfusion (no recognizable color Doppler flow) [28]. Poor perfusion had a positive predictive value of 100% and 81.8% for all necrotizing pneumonias and severe necrotizing pneumonias, respectively. It was also associated with an increased risk of pneumatocele formation and the subsequent requirement for surgical lung resection. However, the absence of color Doppler signals within consolidations has been scarcely studied in adult CAP and warrants further investigation in future research.

Another LUS sign of necrotizing pneumonia in children was the presence of a heterogeneous hypoechoic consolidation containing more hypoechoic confluent lesions [29]. These hypoechoic lesions were thought to correspond to necrotic cavities. Adult studies have shown that the presence of micro-abscesses or hypoechoic areas within consolidations may suggest necrotizing changes, prompting further confirmation with a repetitive CT in suspicious patients (Figure 3) [30,31].

### 2.4. Other Red Flag Signs

For a long time, researchers have seldom incorporated radiologic findings into prognostic models of CAP. The Pneumonia Severity Index (PSI), developed in 1997 to guide hospitalization decisions for CAP in emergency and outpatient settings, included only one radiologic parameter—pleural effusion—among its 20 scoring items [32]. A PSI score of 71–90 corresponds to class III, indicating approximately twice the odds of hospitalization and increased mortality compared to classes I and II. For example, a male patient over 65 years old presenting with pneumonia and pleural effusion would be classified as class III. In such cases, the presence of pleural effusion detected by LUS serves as a red flag. A systematic review reported the incidence of COVID-19-related pleural effusions was 7.3% among 47 observational studies [33]. Pleural effusions were commonly observed in critically ill patients who had Multisystem Inflammatory Syndrome. COVID-19 patients with pleural effusion, compared to patients without pleural effusion, had worse gas exchange and higher mortality in another report [34]. Another study found that CAP patients with pleural effusion were more likely to be older, have comorbid neurological diseases, and have a lasting fever [35]. Notably, pleural effusion is common in patients with heart failure or renal dysfunction and is associated with increased in-hospital mortality [36]. Since PoCUS as an imaging modality has higher diagnostic accuracy than CXR in detecting pleural effusion [37], its presence or new onset should be considered a red flag during CAP management.

The PSI is often considered too complex to calculate, prompting some researchers to propose simplified versions [38]. The CURB-65 score, consisting of confusion, blood urea, respiratory rate, blood pressure, and age over 65, has been widely used for CAP [39]. Developed in 2003, CURB-65 is a simpler tool than PSI; however, it does not incorporate any radiologic findings.

One key clinical implication of PSI and CURB-65 is the prognostic value of comorbidities, physical signs, and laboratory data in CAP. PoCUS serves as a powerful tool to detect occult co-morbidities—such as heart failure, chronic kidney disease, and liver disease—that are incorporated into the PSI score [32]. Laboratory items in PSI and CURB-65 aim to identify signs of sepsis. In this context, PoCUS can help confirm a hyperdynamic left ventricle in patients with tachycardia and/or hypotension [40,41,42], or a hyperdynamic diaphragm in tachypneic patients as a compensatory response [43,44]. Suboptimal diaphragmatic excursion in a patient is an ominous sign, indicating poor respiratory endurance and limited reserve (Figure 4) [45,46,47,48].

In summary, identifying these warning signs with PoCUS can help determine whether a patient with pneumonia should be treated in the hospital or at home. This PoCUS-derived benefit is essential for clinicians practicing HaH care.

## 3. Question 8: Do the Ultrasound Patterns Improve in Accordance with Pneumonia Recovery?

Ultrasound characteristics of the lung, like consolidations, B-lines, pleural effusions, and pleural line disease, typically decrease in size, number, and extent as pneumonia recovers. Serial examinations for both community-acquired and COVID-19 pneumonia indicate that consolidations may regress or resolve; B-lines become lessened; and pleural effusions become decreased and resolve as days and treatment and clinical status become successful [49,50,51]. Most studies confirmed that the course of pneumonia was comparable using X-ray and LUS. In the case of COVID-19 pneumonia, confluent B-lines wane after the acute phase, whereas irregular pleura and subpleural consolidations resolve later [50]. Residual LUS abnormalities can last for months. A study of 96 COVID-19 pneumonia cases found that only 20.8% had complete resolution on LUS at 1 month, which rose to 68.7% at 3 months [51]. Reports in the ICU showed LUS findings were significantly decreased by ICU discharge [52,53]. The use of LUS has been demonstrated to be a powerful tool for monitoring the evolution of COVID-19 during the pandemic [54,55].

In pediatric populations, LUS correlates well with clinical improvement and can reliably monitor disease progression and resolution [56]. In summary, improvement in LUS patterns generally parallels the resolution of clinical symptoms in pneumonia recovery, making lung ultrasound a reliable tool for monitoring disease course and guiding follow-up.

The observation that confluent B-lines and pleural effusions often resolve earlier, whereas subpleural consolidations and pleural line irregularities tend to persist longer—particularly in patients with more severe disease—is clinically meaningful. Recognizing which LUS findings typically improve first is crucial for clinicians’ HaH care, as early reversibility of specific ultrasound features may serve as a reliable indicator of favorable response to antimicrobial therapy. Conversely, an increase in the amount of pleural effusion may indicate a suboptimal or delayed response to therapy. In patients with prolonged complications after pneumonia, such as COVID-19, LUS findings were associated with persistent respiratory symptoms one month after the initial LUS evaluation [57]. It plays an important role in facilitating effective doctor–patient communication within the HaH setting.

Finally, although LUS is a valuable modality for monitoring pneumonia, it has inherent limitations, including incomplete visualization of all pulmonary regions. Therefore, protocols incorporating more than eight scanning zones are recommended to improve coverage, as studies showed that higher acquisition zones rendered higher sensitivity and specificity [58,59]. Moreover, the technique’s operator dependency poses challenges for longitudinal comparisons, particularly when examinations are performed by different clinicians. Questions such as whether “LUS images can be reliably compared across operators” remain insufficiently addressed. To minimize interrater variability, standardized image acquisition protocols and structured training programs for clinicians involved in HaH care should be implemented. Nevertheless, studies evaluating such standardization remain limited in the current literature.

## 4. Question 9: Is Ultrasound Superior to Chest X-Ray for Diagnosing Pneumonia?

The limited sensitivity of chest radiography in the diagnosis of pneumonia has been well described. Lung ultrasound is more sensitive than chest radiography for detecting pneumonia and its complications, and serial ultrasound examinations can accurately track pulmonary reaeration and the effectiveness of treatment [60]. Nevertheless, variations across patient populations and clinical contexts warrant individualized consideration.

### 4.1. Pediatrics

Investigations conducted in the pediatric ICU directly compared the diagnostic performance of LUS and chest radiography (CXR), using thoracic computed tomography (CT) as the reference standard [61,62]. A total of 84 hemithoraces were assessed by all three modalities. For consolidation, CXR demonstrated the sensitivity, specificity, and overall diagnostic accuracy of 38%, 89%, and 49%, respectively, whereas LUS achieved 100%, 78%, and 95%. For interstitial syndrome, the corresponding values were 46%, 80%, and 69% for CXR versus 94%, 93%, and 94% for LUS. For pleural effusion, CXR reported 65%, 81%, and 86%, whereas LUS demonstrated perfect performance at 100%, 100%, and 100%. This observation has been corroborated by another study in the ER setting demonstrating that clinicians can accurately diagnose pneumonia in children and young adults using point-of-care ultrasonography, with high specificity [63]. Another study found that the diagnostic accuracy for childhood pneumonia was greater on LUS than chest x-ray (area under the curve, 0.94 and 0.76, respectively), and LUS missed only 4.5% of the pneumonia cases while chest x-ray missed 21% [64].

Lung ultrasound demonstrates higher diagnostic accuracy for pneumonia in pediatric patients compared to adult patients. In children, pooled sensitivity consistently ranges from 94% to 96% and specificity from 90% to 96%indicating excellent performance [65,66]. In adults, meta-analyses report slightly lower sensitivity (typically 88–94%) and specificity (78–96%), with AUC values around 0.93 [67,68]. Several factors contribute to this difference. Pediatric patients generally have thinner chest walls and less subcutaneous tissue, facilitating better ultrasound penetration and visualization of lung pathology.

Collectively, these findings indicate that LUS has generally demonstrated superior sensitivity to CXR in most major thoracic pathologies, particularly for consolidation and pleural effusion detection. Therefore, strong advocacy has emerged for incorporating ultrasound earlier in the diagnostic imaging algorithm for suspected pneumonia in children [69]. Wherever institutional infrastructure permits, ultrasound may precede, complement, or even replace chest radiography.

### 4.2. Adults

Studies comparing chest radiography and lung ultrasound (LUS) should be contextualized according to specific clinical settings, as adult populations often present with greater complexity than pediatric cases. For instance, in postoperative thoracic surgical patients, chest radiography failed to detect findings observed on ultrasonography in 24% of examinations, and notably missed 60% of pleural effusions that were identified by LUS [70]. In the emergency department setting, LUS markedly reduced diagnostic uncertainty for pneumonia from 73% to 14%, with most of the initial uncertainty attributable to chest radiography findings [71].

A systematic review included 17 studies and found that LUS in the hands of the non-imaging specialists, such as emergency physicians, internal medicine physicians, and intensivists, demonstrated high sensitivities (≥0.91) and specificities (0.57 to 1.00) in diagnosing pneumonia [72]. While chest x-ray interpretation skills vary among non-radiologists, the study observed no substantial difference in diagnostic accuracy between low- and high-performing groups.

When combined with other point-of-care tests, lung ultrasound may enhance clinicians’ confidence in making antibiotic prescribing decisions [73]. This strategy appears particularly promising within the HaH care model.

### 4.3. Special Contexts

A recent study conducted in Viet Nam, a setting with a high incidence of pulmonary tuberculosis, investigated the use of LUS for the diagnosis and monitoring of pneumonia [74]. LUS demonstrated higher sensitivity than chest radiography (CXR)—96.0% versus 82.8%—and comparable specificity of 64.9% versus 54.1%. The moderate specificity of LUS was largely attributable to sonographically similar conditions, particularly pulmonary TB, which was present in 5.1% of patients. Although LUS is highly sensitive for diagnosing pneumonia, its specificity may be limited in TB-endemic regions. Similar limitations were observed in contexts such as the COVID-19 pandemic, where viral pneumonias were more prevalent than those of bacterial origin, and the decision must be individualized [75].

Patients with obesity, thick chest walls, subcutaneous emphysema, and restricted chest wall access (from dressings, prosthetic material, or dermatologic conditions) are prone to reduced diagnostic accuracy with LUS for pneumonia. In these populations, ultrasound beam penetration and transmission are impaired, limiting visualization of the lung parenchyma and hindering detection of consolidations. The American College of Radiology specifically notes that LUS has limited utility in such contexts [76].

## 5. Question 10: Does Ultrasonography Lead to Overdiagnosis of Pneumonia?

Lung ultrasound has consistently demonstrated higher sensitivity than chest radiography in various clinical applications, frequently detecting radiographically occult abnormalities that might otherwise go unrecognized. However, a highly sensitive diagnostic tool may also detect minor or clinically insignificant pathological findings. It remains uncertain whether all patients with suspected lower respiratory tract infection and LUS-detected but radiographically occult consolidations truly require antibiotic therapy [77].

Overdiagnosis of pneumonia leads to concerns of unnecessary antibiotics and antimicrobial resistance [78]. The potential for overtreatment of such radio-occult conditions warrants systematic evaluation through dedicated clinical trials. However, challenges of the robustness of clinical trials for PoCUS remain. First, gold standards, such as retrospective medical record review or expert opinion, used to investigate the diagnostic accuracy of LUS for pneumonia may not always be appropriate. This limitation reflects the fact that the diagnosis of pneumonia is not solely image-dependent. Second, given the high operator-dependent variability of PoCUS, pre-study hands-on workshops are recommended to standardize sonographic competency and minimize bias [79].

A recommended approach to minimize the overdiagnosis of pneumonia by LUS is to integrate ultrasound findings with the clinical context and complementary diagnostic information from other modalities. It is important to emphasize that ultrasound assessment should not be confined to the lungs; for instance, in reports of decision-making for COVID-19 patients, age, oxygen saturation, sonographic measurement of inferior vena cava diameter, pleural space, and pericardium were also incorporated [14,15]. For example, understanding the geographic vaccination [80,81] and seasonal patterns of infectious diseases is paramount [82,83], and biomarkers are associated with treatment outcomes [84,85]. Even as LUS-based quantification for pneumonia diagnosis evolves [86], each patient warrants a holistic biopsychosocial approach. We appreciate the American College of Physicians guideline’s concept of “parallel use,” in which PoCUS complements rather than replaces standard diagnostic pathways to enhance diagnostic accuracy, while “replacement” refers to substituting one or more tests entirely with PoCUS [87]. POCUS used in parallel with standard diagnostics can improve sensitivity and reduce missed cases, representing the most common clinical scenario. Yet, most studies have assessed POCUS as a replacement for chest radiography, citing its simplicity, safety, and potential accuracy advantages [88]. We advocate that future researchers avoid using replacement or head-to-head designs when evaluating the role of PoCUS. Emphasizing its parallel, integrative use will yield more clinically meaningful and generalizable insights.

## 6. Conclusions

Effective diagnosis and management of pneumonia in hospital-at-home programs require clinicians to master the identification of sonographic patterns of pneumonia, supported by the appropriate selection of equipment and scanning techniques. In the first part of our review, we summarize current practices for diagnosing pneumonia by LUS, focusing on different protocols in the literature. In the second part of our review, we summarize the evidence supporting the repetitive application of PoCUS throughout the clinical course of pneumonia. Recognizing differential diagnoses of LUS patterns, along with awareness of potential pitfalls and confounders, is essential for improving diagnostic accuracy and delivering person-centered care in HaH programs. In the third and final part of the review, we present current evidence showing that LUS findings can provide prognostic insights during the early phase of pneumonia and exhibit synchrony with disease progression or recovery. Finally, we emphasize that clinicians should not rely solely on LUS findings, while further multicenter validation is warranted, when evaluating patients with pneumonia in home-based settings, as we are ultimately treating patients—not diseases.

In the HaH setting, LUS offers a practical, radiation-free, and repeatable imaging modality that aligns well with the logistical and safety constraints of home-based care [1,2,3,4,5,6]. When integrated with structured clinical assessment and longitudinal monitoring, LUS may support timely escalation or de-escalation decisions without compromising patient-centeredness. Future research should focus on implementation frameworks [89,90], operator training [91,92], and outcome-driven validation [93] to define the optimal role of LUS in HaH pneumonia management. In the future, equipping the primary healthcare workforce, including family physicians, with competencies for independent practice will be essential to enhance the attractiveness, professional identity, and prestige of home-based medical careers [94].

## Figures and Tables

**Figure 1 diagnostics-16-00192-f001:**
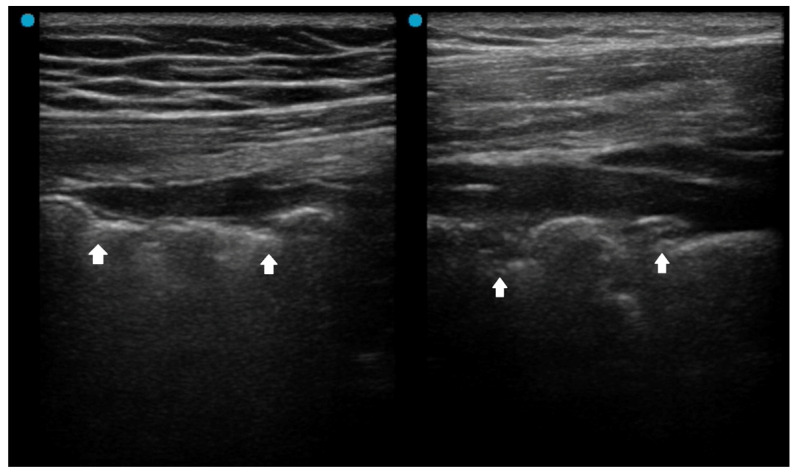
Pleural irregularity and thickening (arrows) in a patient with acute respiratory distress syndrome.

**Figure 2 diagnostics-16-00192-f002:**
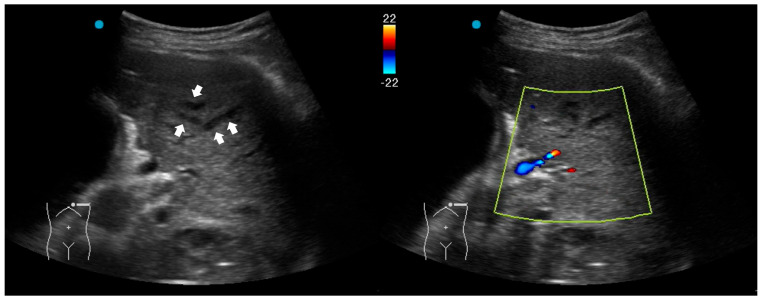
Fluid bronchogram (arrows) in a patient with left upper lobe obstructive pneumonia. Vessels and fluid-filled bronchi can be differentiated with color Doppler (the light yellow box indicates the region of interest box).

**Figure 3 diagnostics-16-00192-f003:**
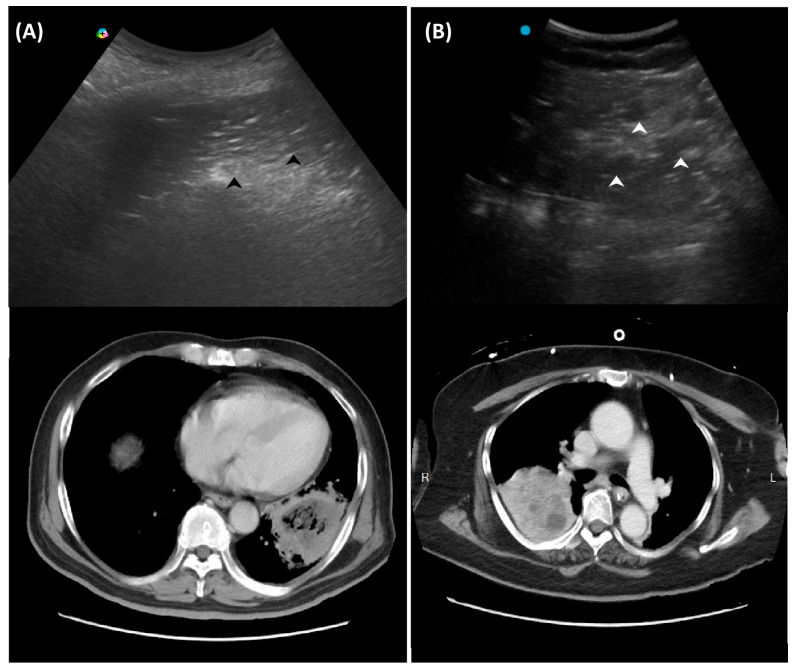
Ultrasound findings in necrotizing pneumonia and corresponding computed tomography images: (**A**) Micro-abscesses (black arrowheads); (**B**) Hypoechoic lesions within consolidations (white arrowheads).

**Figure 4 diagnostics-16-00192-f004:**
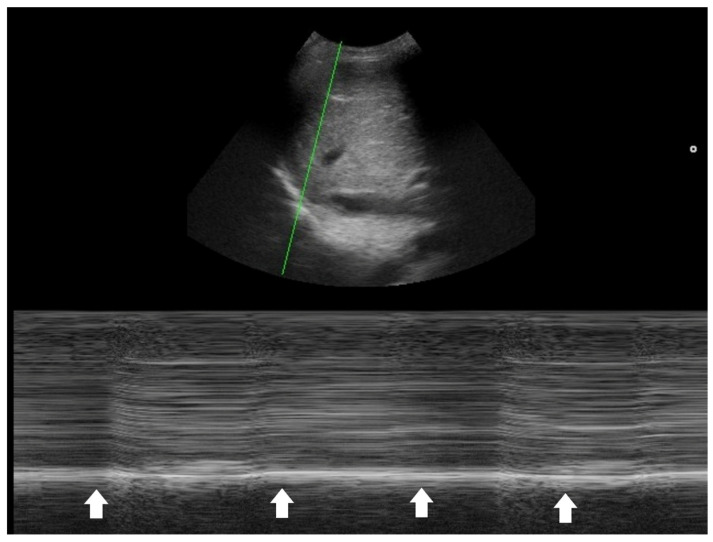
M-mode tracing of the diaphragm (green line indicates the scanning line) showed minimal respiratory excursion (arrows), indicating poor diaphragm function.

**Table 1 diagnostics-16-00192-t001:** Ten essential questions to address before using point-of-care ultrasonography to diagnose and manage pneumonia in the hospital-at-home model.

What ultrasound techniques are essential for diagnosing pneumonia?
2. What are the ultrasound patterns associated with pneumonia?
3. Do different settings or etiologies of pneumonia influence the diagnostic accuracy of ultrasonography?
4. Do pulmonary comorbidities affect the accuracy of ultrasound diagnosis for pneumonia?
5. Do other differential diagnoses mimic the ultrasound patterns of pneumonia?
6. Do ultrasound findings correlate with pneumonia severity?
7. Do initial ultrasound findings associated with pneumonia hold prognostic value?
8. Do the ultrasound patterns improve in accordance with pneumonia recovery?
9. Is ultrasound superior to chest x-ray for diagnosing pneumonia?
10. Does ultrasonography lead to overdiagnosis of pneumonia?

**Table 2 diagnostics-16-00192-t002:** Red flag signs on ultrasound for patients with pneumonia suspicious for complications.

Red Flag Signs	Implications
Pleural line abnormality (thickened, irregular)	Possible ARDS
Absence or reduction of pleural sliding (gliding sign)	Possible ARDS
‘Spared areas’ within confluent B-lines	Possible ARDS
Nonhomogeneous distribution of B-lines	Possible ARDS
Fluid (liquid) bronchogram	Post-obstructive pneumonia
Absence of color Doppler signals within consolidation (poor perfusion)	Necrotizing pneumonia
Hypoechoic lesions or microabscesses within consolidations	Necrotizing pneumonia
Pleural effusion	Possible complicated pneumonia
Hyperdynamic left ventricle	Sepsis
Suboptimal diaphragm excursion	Poor respiratory strength or endurance

## Data Availability

Not applicable.

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
