# Peer review of "Ten Questions on Using Lung Ultrasonography to Diagnose and Manage Pneumonia in Hospital-at-Home Model: Part III—Synchronicity and Foresight"

_diagnostics, 2026, doi:10.3390/diagnostics16020192_

Round 1
Reviewer 1 Report
Comments and Suggestions for Authors
This manuscript represents the third installment of a three-part series on the use of lung ultrasonography (LUS) for diagnosing and managing pneumonia in the hospital-at-home (HaH) model. Building on the earlier discussions in Parts I and II—which focused on technical aspects and diagnostic confounders—this paper centers on the clinical and interpretive dimensions, namely the prognostic value of LUS findings, the synchronicity between ultrasound patterns and disease recovery, its comparative diagnostic performance against chest radiography, and the potential issue of overdiagnosis. The overall structure is clear, the thematic progression coherent, and the discussion well supported by recent literature. It continues the series’ logical development and provides a complete conceptual framework for clinicians applying LUS in home-based care settings. The writing demonstrates solid organization and reflects a mature understanding of both the technology and its clinical context.
The manuscript’s strengths lie in its clarity, logical structure, and clinically grounded argumentation. The paper effectively articulates the concepts of “synchronicity” and “foresight,” showing how LUS can serve as both a diagnostic and monitoring tool. Each section follows the “Ten Questions” framework, providing continuity with the prior papers and allowing readers to easily navigate the discussion. The figures and tables are well used, and the literature cited is current and relevant, incorporating multiple recent studies on COVID-19 and community-acquired pneumonia. The conclusions are balanced and clinically sound, emphasizing that LUS should complement rather than replace standard diagnostic pathways—a prudent and well-supported stance.
The revisions needed are primarily minor, concerning technical presentation and wording. The most significant issue is Reference 19, which currently appears only as “第二篇” (“the second paper”) and must be replaced with the full citation as follows:
Hsu N.C., Lin Y.F., Tsai H.B., Liao C., Hsu C.H. Ten Questions on Using Lung Ultrasonography to Diagnose and Manage Pneumonia in the Hospital-at-Home Model: Part II—Confounders and Mimickers. Diagnostics 2025, 15(10), 1200. https://doi.org/10.3390/diagnostics15101200
.
The authors should also review the entire reference list for punctuation and stylistic consistency—some entries contain double periods, missing DOIs, or inconsistent italics for journal titles.
Minor typographical and grammatical issues are also present. For example, “reg flag signs” should read “red flag signs,” “sough to differentiate” should be “sought to differentiate,” and “summarises” should be standardized to the American form “summarizes.” These are small but worthwhile corrections to ensure language accuracy and consistency.
In a few places, the tone is somewhat too assertive and would benefit from softening to maintain an objective and evidence-based voice. For instance, the statement “absence or reduction of pleural gliding was observed in 100% of ARDS and 0% of APE” could be revised to “was consistently reported in ARDS and rarely observed in APE.” Similarly, “LUS consistently outperformed CXR across all major thoracic pathologies” could be modified to “LUS has generally demonstrated superior sensitivity to CXR in most major thoracic pathologies.” In the conclusion, “clinicians should not rely solely on LUS” could be expanded to “clinicians should not rely solely on LUS, while further multicenter validation is warranted.” These refinements adjust tone without altering meaning and will make the manuscript appear more balanced and scientifically cautious.
Overall, this is a well-written, comprehensive, and clinically meaningful review that successfully extends the authors’ earlier work from diagnostic principles to prognostic and decision-making perspectives. The paper is suitable for publication after minor revision, primarily addressing the reference formatting, typographical errors, and a few overly definitive statements. Once these adjustments are made, the manuscript will meet the standard for publication in Diagnostics.
Author Response
Dear Reviewer,
Thank you for positive feedback on our manuscript, and also giving us very useful recommendations in improvement. We are pleased to response with our revisions.
- Reference 19: We have added the part II paper’s full citation as “Hsu N.C., Lin Y.F., Tsai H.B., Liao C., Hsu C.H. Ten Questions on Using Lung Ultrasonography to Diagnose and Manage Pneumonia in the Hospital-at-Home Model: Part II—Confounders and Mimickers. Diagnostics 2025, 15(10), 1200. https://doi.org/10.3390/diagnostics15101200”
- We also revise the entire reference list for punctuation and stylistic consistency as your suggestions. All corrections are labeled with red colors.
- Minor typographical and grammatical issues as you noticed: “reg flag signs” has been corrected to “red flag signs,” “sough to differentiate” to “sought to differentiate,” and “summarises” to “summarizes.”All these corrections are labeled with red colors.
- We are also thankful to your recommendations on sentence revisions. The statement “absence or reduction of pleural gliding was observed in 100% of ARDS and 0% of APE” has been revised to “absence or reduction of pleural gliding was consistently reported in ARDS and rarely observed in APE.”
- “LUS consistently outperformed CXR across all major thoracic pathologies” has been revised to “LUS has generally demonstrated superior sensitivity to CXR in most major thoracic pathologies”
- The conclusion remark “clinicians should not rely solely on LUS…” has been revised to “clinicians should not rely solely on LUS findings, while further multicenter validation is warranted, when evaluating patients with pneumonia in home-based settings, as we are ultimately treating patients—not diseases.”
Finally, we appreciate your support and are grateful for your positive assessment that our paper is suitable for publication pending minor revisions.
Reviewer 2 Report
Comments and Suggestions for Authors
This is the third of a review series on lung ultrasound´s role in diagnosing community acquired pneumonia, an interesting and important topic. The authors´ first paper began with an introduction to the hospital-at-home model, which through portable diagnostics, can deliver acute care in patients’ homes. As this is the final paper of the series, it should circle back to the hospital-at-home approach and reinforce the role of LUS in this setting. The authors can do so in greater detail, in the “conclusion" section. Regarding the rest of the manuscript, some specific comments are:
Line 18: This is the first time you are mentioning lung ultrasound on your abstract. As you did above, first mention the full term with the abbreviation in a parenthesis.
Line 39: Do these references indicate an adaptation of the HaH model, to fit into the respective healthcare systems of each geographical region you are mentioning? Or are you referring to the model’s popularity in both west and east? If it is the latter, “adoption” should be the correct term.
Line 51: Replace “mitigate” with “mitigating”.
Lines 61-73: These two paragraphs are not consistent as they are. In the first paragraph, the main point is that CT in the ER offers stronger diagnostic consistency regarding CAP, as opposed to chest X-ray, as well as some prognostic value. In the second paragraph, you very correctly make the transition into PoCUS´s prognostic value. Firstly, the point about repeatability in the second paragraph should be rewritten, to highlight that while both diagnostic tests are delivered once (as you mention in both paragraphs) PoCUS offers some advantages when compared to CT in the ER for CAP. Secondly, the point about follow-up either does not belong in this part of the text as your main point is the prognostic value of the initial findings, or it should be rewritten as an “add-on” feature of US. Finally, a short sentence should be added at the end of line 73, that introduces Table 2.
Line 76: “…pneumonia suspicious for complications” would better serve your point.
Lines 77-121: This section outlines US findings that hold diagnostic value for ARDS or other complications. While your point is to link sonographic findings to more severe illness and therefore worse prognosis, it is not clear in the text that this is the main goal. Please elaborate. Lines 141-144 can be used as an example, as in this section, your point is crystal clear: PoCUS is a better diagnostic tool than X-ray to capture the presence of pleural effusion, a finding with clear and documented prognostic value.
Line 80: Replace “sough” with “sought”.
Line 86: Replace “ALI” with “acute lung injury”.
Line 136: Replace “and” with “that”.
Lines 146-148: This is a very interesting point and of great importance to your paper. However, CURB65 is not recent, as stated in line 146. Furthermore, many researchers challenge its prognostic accuracy (example: 10.1016/S2213-2600(25)00124-9 ), despite it being widely used in international ER settings. This shift in trusting the CURB65, serves your point about incorporating radiologic findings in CAP approach and your paper would benefit from an extension of this section.
Lines 69-70: Use coherent tense. Either waned/resolved or wane/resolve.
Line 211: Remove “which”.
Line 212: This is not the first time you have mentioned LUS. Write the full term the first time LUS is mentioned and use “LUS” every time thereafter.
Line 219: Add “setting” after “ER”. Same for “CXR” and “CT” in Line 213.
Line 284: What are these gold standards you are referring to? Please elaborate.
Line 292: Age and oxygen saturation are indeed metrics. “Inferior vena cava” is not a standalone metric to be incorporated into decision making. Are you referring to sonographic assessment of the IVC? If so, it should be mentioned clearly, for IVC and for other possible metrics.
Author Response
Dear reviewer,
Thank you for your positive feedback on our manuscript and for providing highly useful recommendations for improvement. We are pleased to respond with our revised submission. All the corrections are marked with red colors.
- In Line 18 when the first time we mentioned lung ultrasound, we add full name “lung ultrasound (LUS)”
- Line 39: Thanks for your comments, we revised “adapted” to “adopted”.
- Line 51: As your suggestion, we have replaced “mitigate” with “mitigating”.
- Lines 61-73: We agree that the point about repeatability of PoCUS, but not CT, should be rewritten, to highlight that while both diagnostic tests are delivered once (as you mention in both paragraphs). In the revised manuscript we stated that “PoCUS offers advantages in safety and repeatability that are not typically achievable with CT.” We also remove the mentioning of “follow-up” as your suggestion. Regarding Table 2, a brief explanatory sentence already exists at line 75, and we have added spacing between the table caption and the text to avoid confusion.
- Line 76 (78 in the revised manuscript): “Red flag signs on ultrasound for patients with pneumonia suspicious for complications.”
- Lines 77-121: As your suggestion, we elaborate our content by adding “PoCUS performed early in the course of pneumonia may provide valuable infor-mation regarding disease severity and the potential for complications.” (Line 75-76), and later “In summary, identifying these warning signs with PoCUS can help determine whether a patient with pneumonia should be treated in the hospital or at home. This PoCUS-derived benefit is essential for clinicians practicing HaH care.” (Line 161-163).
- Line 80 (83 in revised manuscript): We have replaced “sough” with “sought”.
- Line 86 (89 in revised manuscript): We have replaced “ALI” with “acute lung injury”.
- Line 136 (139 in revised manuscript): We have replaced “and” with “that”.
- Lines 146-148: We removed “recently” statement for CURB65. We agree that incorporating radiologic findings in CAP approach to CURB65 would be beneficial.
- Lines 69-70 (be 175-176 in the revised): We have used coherent tense, wane/resolve.
- Line 211 (217 in revised manuscript): We have removed “which”.
- Line 212 (218 in revised manuscript): We use “LUS” in the revised manuscript.
- Line 219 (225 in revised manuscript): We added “setting” after “ER”.
- Line 284: We elaborate our statements with “First, gold standards, such as retrospective medical record review or expert opinion, used to investigate diagnostic accuracy of LUS for pneumonia may not always be ap-propriate. This limitation reflects the fact that the diagnosis of pneumonia is not solely image-dependent.”
- Line 292 (300 in revised manuscript): “Inferior vena cava” has been revised to “sonographic measurement of inferior vena cava diameter”.
We appreciate your detailed and careful review of our manuscript. It improves much in clarity after your recommendations. Thanks.
Reviewer 3 Report
Comments and Suggestions for Authors
Congratulation. Very important work. Several minor remarks:
In the abstract should the abbreviation LUS outlined
Some hyphen should be omitted e.g. pop-ulation
Fig 2 Legend. the fuidbronchogram could also be a vein, lacking wall, could differentiated with color Doppler
Fig 3 Put A, B in the picture
Line 146 explain CURB-66
Author Response
Dear reviewer,
We appreciate much your review and comments on our manuscript. We are pleased to respond to your comments with revisions.
- In the abstract should the abbreviation LUS outlined.
Response: Yes, it has been added with lung ultrasound (LUS).
- Some hyphen should be omitted e.g. pop-ulation
Response: All hyphen have been checked and revised. Thank you.
- Fig 2 Legend. the fuidbronchogram could also be a vein, lacking wall, could differentiated with color Doppler
Response: Thanks. We add this important remark to the legend and it will be like: “Figure 2. Fluid bronchogram (arrows) in a patient with left upper lobe obstructive pneumonia. Vessels and fluid-filled bronchi can be differentiated with color Doppler.”
- Fig 3 Put A, B in the picture
Response: Thanks. We have revised the photo file of Figure 3 with (A) and (B).
- Line 146 explain CURB-66
Response: Thank you. We are pleased to briefly introduce CURB-65 in the revised sentence: “The CURB-65 score, consisting of confusion, blood urea, respiratory rate, blood pressure, and age over 65, has been widely used for CAP”
We hope our revisions meet your standards. Thank you.
Round 2
Reviewer 2 Report
Comments and Suggestions for Authors
The authors have adequately addressed the individual comments on the original manuscript, and the overall quality of the revised version is now suitable for publication. However, one minor issue remains. As the final paper in this review series, the manuscript should more strongly reinforce the hospital-at-home framework. In its current form, the authors’ original concept and overall trajectory are not yet sufficiently substantiated by this concluding paper. The authors are therefore encouraged to further expand and strengthen the Conclusions section to more clearly promote and contextualize the LUS approach they advocate. When they do so, the manuscript will be ready to publish.
Author Response
Thank you for this suggestion. We have revised the Conclusion section to better reflect the role of lung ultrasound in hospital-at-home care.
In the original manuscript, we have summarized the goal of part I to Part III of our review. We further added the following statements with new references:
“In the HaH setting, LUS offers a practical, radiation-free, and repeatable imaging modality that aligns well with the logistical and safety constraints of home-based care [1-6]. When integrated with structured clinical assessment and longitudinal monitoring, LUS may support timely escalation or de-escalation decisions without compromising patient-centeredness. Future research should focus on implementation frameworks [91, 92], operator training [93, 94], and outcome-driven validation [95] to define the optimal role of LUS in HaH pneumonia management. In the future, equipping the primary healthcare workforce, including family physicians, with competencies for independent practice will be essential to enhance the attractiveness, professional identity, and pres-tige of home-based medical careers [96].”
We hope these revisions meet your expectations. Thank you.